# Sustainable and conventional banking in Europe

**María del Carmen Valls Martínez** *, Salvador Cruz Rambaud, Isabel María Parra Oller

Economics and Business Department, University of Almería, Almería, Spain

* mcvalls@ual.es

## Abstract

At the end of the 20th century a new banking model, the so-called ethical banking, emerged becoming the maximum exponent of a socially responsible investment. The financial crisis in 2008 led to a distrust of the conventional financial system and consequently investors began to look with interest this new banking, which only invests in ethical activities and products, with social and environmental criteria, total transparency and a democratic management. The aim of this article is to analyze the economic structure of ethical banking, compared to that of conventional banking, by paying attention to its liquidity, coverage and solvency. Specifically, We compare the financial statements of Triodos Bank, the main European ethical bank belonging to the Global Alliance for Banking on Values, with two of the main conventional banks of each of the five countries in Europe in which it operates. To do this, we apply a financial and economic analysis to the period from 2015 to 2018, the means difference test and analysis of variance on an array of financial ratios and, finally, probit regressions. The results reveal that ethical banking is growing more than conventional banking and it presents greater liquidity and solvency, although, in general terms, its profitability is not higher. In conclusion, both savers and investors have guarantees that their savings are invested not only in a responsible but also in a confident way in ethical banking.

## Introduction

As officially recognized by economic agents, the Lehman Brothers' bankruptcy in 2008 marked the beginning of the financial crisis in the main developed countries, whose effects remain until today. Banks caused the financial crisis due to their irresponsible lending policies which led to a reckless accumulation of toxic assets. This led to a situation that no person could anticipate, since banks were subject to strict regulations and projected an image of security and risk control. As a consequence, a good number of governments had to take measures to guarantee savings, through financial aids and nationalizations. Moreover, a more strict regulation was developed; in this way, the Basel Committee applied a set of measures, between 2012 and 2019, for the management of systematic risk in the European banks [1]. However, banks did not act illegally, but they exhibited a lack of morality. In fact, the causes of such severe crisis were not only economic, but also ethical; in this sense, we can consider individual moral failures, ethical failure related to management or governance, and social ethics failure [2].

**Competing interests:** The authors have declared that no competing interest exist.

Therefore, a regulation with greater constraints will not be enough to avoid, in the future, the same problems of the past. It would be necessary that there was a change of thought towards a virtue-based ethic [3]. If we want to enhance an attitude towards money and finance in a more humanistic direction, we will have to increase a sound education and practices in social banking and finance [4].

Having said that, in last decades, several small banks have chosen a different business model, where the profit is considered as a means to an end, and not as an end itself, as we are going to see in this article. They are the so-called ethical banks, which are proactive in creating a sustainable world and which should be an inspiration to the banking sector [5].

Ethics has been linked to banking since its birth. The first banks arose in Italy in the Renaissance, where Catholicism was the prevailing moral, so that usury was not allowed and the access to credit of most disadvantaged people was favoured. The "Montes de Piedad", created by religious orders, were the basis for the first saving banks in the first half of the 19th century and had a clear social vocation. Likewise, the cooperative banks, which also emerged in the 19th century, in the midst of the Industrial Revolution, had the mission of financing local projects and favouring the saving of working classes. However, the first ethical investment fund, the Pax World Fund, was created in 1971 on the occasion of the pacifist movement that emerged in the US in the wake of the Vietnam War. People rejected conventional banking to deposit their savings, since it financed arms companies interested in prolonging the conflict. It was in the three last decades of the 20th century when the first properly called ethical banks were created: the South Shore Bank in US (1973), Triodos Bank in Holland (1980), The Cooperative Bank in England (1992), Banca Popolare Etica in Italy (1995) and JAK Medlemsbank in Sweden (1997) [6].

Ethical banking emerged linked to the concepts of socially responsible investment (SRI) and corporate social responsibility (CSR). The SRI gives a social dimension to the usual economic criteria of banks, whilst banks must act with CSR so that their activity has a positive influence in the society [7]. Since the 2008 financial crisis, these banks that consider people before economic profit have not stopped gaining recognition and grow unceasingly in strength and number [8]. Nevertheless, nowadays, ethical banking remains unknown to conventional banking, scholars and, of course, the public in general [9].

We can highlight several factors that contribute to the success of this kind of banks [9–11]. First, we can highlight the distrust towards conventional banks since the beginning of the financial crisis as they have been accused of unethicalpractices. Moreover, numerous financial entities have experienced bailouts. As a result, the economic growth has been limited and depositors and other stakeholders have been damaged [12]. Therefore, banks are required to operate in a more responsible way. Second, the wide and increasing information about the social and environmental problems caused by the economic progress has enhanced the interest in sustainability, by propitiating the transformation of "monetary centralized economics" into "socio-environmental centralized economics". Namely, the change of profit maximization to sustainable socio-environmental development into banks operations, where not only economic but also social and environmental costs are considered, are assessed [13]. To sum up, the attractive values that ethical bank upholds and its ability to innovate products and services that consider social and environmental factors, are the main factors that explain its evolution. Despite this, academic research on ethical banking is still in an incipient stage, which is probably due to this business model represents only a small niche within the total banking system [14–15].

SRI has experienced a rapid growth in the last years and is becoming a mainstream kind of investment [16]. The term SRI comes up in the United States at the middle of the 20th century, and was mainly linked to the repulse of certain religious groups that their money was used in

unethical investments, such as gamble, tobacco or alcohol. A decade later, the movement was enlarged and, as said above, people were opposed to deposit their money in banks that financed companies whose activities were related to the Vietnam War and the Apartheid in South Africa [17].

The term SRI is used to refer to the investment that takes into account not only economic criteria, such as profitability and risk, but also environmental, social and governance issues, which are known as ESG criteria; namely, the funds have to be always invested according to ethical and financial criteria jointly [11,18]. Thus, investors who want to invest in a responsible way have four options: first, they could deposit their savings in bank accounts funding only socially responsible projects, such as ethical banks; second and third, they could become shareholders of socially responsible companies, both directly and through mutual funds; fourth, they could invest to get a management position in a company and, consequently, to take socially responsible decisions [19]. An investment can be classified as SRI by using positive or negative screens, which enable to reject undesirable activities (e.g., nuclear power, arms productions or gambling) and to concentrate funding into sustainable sectors (e.g., organic agriculture, renewable energy or recycling industry) [9, 19].

In fact, today an increasing number of shareholders consider social and environmental criteria, besides financial criteria, in their investment making decisions [16]. We can distinguish two main groups of shareholders: institutional and retail investors, who have the information provided by rating organizations specialized in SRI and sustainability indices, such as FTSE4-Good index and DOW Jones sustainability index. The retail sector will become the cornerstone of sustainable finances, since from 2013 to 2017 the demand in the retail sector increased by over 800%, even though almost 70% of SRI assets belonging to institutional investors in 2017, and European households savings represented over 40% of total financial assets in the EU [20]. Nevertheless, the degree of SRI still remains small compared to overall investment, maybe because investors have a low level of knowledge on social investing [21]. In other cases, investors opt to put only a small portion of their savings into SRI funds, as a means to alleviate their consciences and legitimize the rest of their conventional investment [19].

On the other hand, CSR can be considered as the underlying framework for sustainable finance [11]. The World Bank defines CSR as "the commitment of business to contribute to sustainable economic development, working with employees, their families, the local community and society at large to improve quality of life, in ways that are both good for business and good for development" [22]. Thus, the old concept proposed by Friedman [23], which argued that the only concern of companies is profit maximization, nowadays has been changed by the Stakeholder Theory [24], by considering that, besides shareholders, there are other parts involved in corporate activities. Therefore, companies that experience a good social and environmental behavior will have a competitive advantage over the companies that do not make such efforts [11].

Responsibility is a twofold concept, because implies anticipating the consequences of actions carried out (responsiveness) and reporting back of them (accountability) [25]. Companies have to be socially responsible for both moral (by compensating for social and environmental damage) and practical (economic profit) reasons [26]. In this way, several studies on the banking sector have found a positive link between social and financial performance or a non-statistically significant link, but never a negative relationship, at least in the long term [1,27,28].

Financial ethics must promote real or productive economic activities and get the common, not the individual welfare [2]. Accordingly, the Goldman Rule "pursue profitable opportunities regardless the effects on others" is not applicable to ethical banking [29]. The financial sector, that includes banks, pension plans and other financial institutions, is a key piece to change

economic activity towards sustainable development, since it multiplies the capital and manages the risk. Therefore, ethical investment to promote sustainability should no longer be a discretionary option for financial intermediaries [30]. In this way, capital markets are important drivers for implementing socially and environmentally responsible policies by the companies and, specially, banks which have a huge impact on society. For example, banks can offer price differentiation, by charging higher interest rate to companies with lower CSR performance. Moreover, they can promote sustainable products or channel funds to certain sustainable activities [11,31]. In fact, the banking socio-environmental responsibility goes beyond its internal operations (e.g., paperless systems, reduced carbon footprint), since through its external operations, by distributing resources, it can contribute in a substantial way to the society and environment [13] and influence the pace and direction of economic growth [32]. Therefore, banking can contribute to improve the stability of the financial system [33].

It is true that the global banking sector has begun to take the first steps since the beginning of the financial crisis. In this sense, we could mention the growing number of CSR disclosures, the inclusion of responsible banking products (e.g., ethical pension plans, ethical investment funds) or the sponsorship and donation activities. However, there is still a long way to produce a total change of policies [1,13], until achieving the category of ethical banking.

Our findings show that in last years, conventional banking has decreased o experienced a low increase; on the contrary, ethical banking has had a high growth that has been financed by client funding, which implies that this kind of banking is more independent of the financial system. Moreover, on the one hand, profitability of ethical banking is not higher than that of conventional banking. But, on the other hand, liquidity, solvency and guarantee is noticeably higher in ethical banking, which implies a lower risk; the main raison of this is that ethical banking develops its operations within the real economy and it refrains from speculation.

## Foundations and research hypotheses

Ethical banking follows a socio-economic model, by contributing to sustainable (social, environmental and economic) development [9, 34]. Without economic profitability, banks would not be sustainable over time and, without social-environmental dimension, they would be merely a bank [18]. Therefore, ethical banks are conditioned not only by the profit, but also by people and the planet [4]. The three aspects of ethics in banking are: integrity, responsibility and affinity [35]. Thus, professional bankers must act with integrity in order to generate trust among depositors, which is only possible if there is an additional self-regulation to external laws. Responsibility implies to execute an appropriate loan policy and to avoid financial exclusion of specific segments of the society. Finally, affinity leads to a closer relationship between depositors and borrowers, which are only possible by providing a high level of transparency [12, 13].

There is another terminology to refer to ethical banking, which is gaining acceptance, such as banking on values, sustainable banking or alternative banking [15]. However, the concept of social banking is wider, since it includes not only ethical banks, but also savings and cooperative banks [7]. Effectively, savings and cooperative banks share social characteristics with ethical banking and, for this reason, they are jointly included in the concept of social banking, but they are not properly ethical banks, since their whole investment is not guided by social-environmental criteria and this is a necessary requirement to be considered as ethical banking. Moreover, it is important to differentiate ethical banks from poverty-alleviation banks [5]: the first ones are aimed to gather customer deposits to finance cultural, social and ecological projects, whilst the second ones aim the economic development for the low-income population as well as the community development in marginalized areas, generally through microfinance.

The transparency of information and allocation of assets, against banking opacity, is a differential factor of ethical banking versus conventional banking, so that depositors know where their money has been lent [34]. In fact, an important share of population is willing to forgo a significant part of their personal financial returns to favor social outcomes, namely, they are socially-minded depositors [36]. It is usual that ethical banks lend to projects non well-understood by conventional banks, and sometimes with reduced rate of interest [37]. All this can be considered as a responsible sourcing and distribution of funds [38]. Actually, sustainable investments and lending practices improve the quality of life for the greater possible amount of people and their positive effects remain and multiply over time [4].

Speculative transactions, which were the main problem for conventional banks during the financial crisis, are refrained in ethical banks, by instead focusing on activities with a positive impact on real economy; this implies the strengthening of long-term instead short-term relations [8–10]. But, there exist more differences between the two banking models. In effect, ethical banks meet certain needs unsatisfied by the banking sector, by developing an active role against the financial exclusion through financing of companies and projects belonging to the social economy [7]. Thereby, funding people and activities unserved by other banks, leads ethical banking towards a specialization in specific sectors and, as a consequence, they can offer pioneering products [5], e.g., mortgages with interest rates linked to the energy rating of the property. Moreover, the ethical banks' depositors have a profile with a social vocation and knowledge about electronic banking, since they have to operate through internet, given the low number of branches at their disposition, since ethical banks have a limited physical presence [7]. Moreover, frequently founders of ethical banks are no bankers, but social organizations or socially driven individuals, for example. Finally, they are profit-making, not profit-maximizing banks, in monetary terms [5].

Accordingly, it is clear that there exist two models of banking business: conventional banks, which had to make changes in their behavior after the financial crisis, and ethical banks, whose behavior has scarcely been altered by the new financial context [39]. Table 1 shows the main differences between ethical and conventional banking.

There is no hallmark or certificate to recognize a bank as ethical and the affiliation to the ethical banking movement is voluntary [15]. Most of them are integrated into the Global Alliance for Banking on Values (GABV), an independent network founded in 2009 that comprises 54 financial entities (banks, banking cooperatives and credit unions) operating all around the world. The GABV is based on the following five principles: the triple bottom line approach of

**Table 1. Main differences between ethical and conventional banking.**

| Feature | Ethical banking | Conventional banking |
|---|---|---|
| Objective | Triple bottom line: social, environmental and economic | Profit-maximizing to reward shareholders |
| Profit | A means to and end | An end in itself |
| Investment object | Only on real economy | Speculative transactions |
| Investment criteria | Positive screen (environment, employment creation, culture, etc.) and negative screen (armament, polluting companies, child exploitation, etc.) | Profitability and risk |
| Loan policy | Avoid financial exclusion | Exclusion of specific segments of the society |
| Transparency | Total transparency of information and allocation of assets | Lack of transparency |
| Geographical distribution | Low number of branches | High number of branches |

Source: Made by authors.

people, planet and profit, serving the real economy and the community, long-term relationship with clients, long-term resiliency and transparency [40].

Ethical banking is, in essence, a European phenomenon [9]. Table 2 shows the main figures of the European banking entities included in the GABV for the 2015–2018 period. We can observe that all of them are of a small size, although both Triodos Bank and Crédit Coopératif are noticeably higher than the rest. In only three years, the size, measured by total assets, has increased over 37%; the volume of loans and client funding have experienced similar growth, with over 37% and 32%, respectively. Triodos Bank, in particular, has increased its total assets by 38.93%, loans by 46.37% and client funding by 37.76%. It is important to highlight the increase in equity, over 37% on average, and 52.17% in the case of Triodos. However, the total revenue has had a lower growth; but, in any case, the data are fine. These financial entities have more than 1.75 million of clients in Europe and 5,000 employees. If the clients of this kind of banking are increasing in Europe, we will have to think that they are satisfied with the quality of the received services. Moreover, if the clients are more difficult to be satisfied in regions with high economic level [41], such as Europe, we will have to conclude that the services offered by ethical banks are of high quality.

Triodos, registered in the Netherlands, is the bank with a higher number of employees and it will be our case of study and the main basis on which making the comparison with conventional banking. The mission of Triodos Bank is to create a society where the quality of life of all people and the environment are protected, placing human dignity at the center of its activities. Thus, it focuses its lending exclusively on projects socially and environmentally beneficial, besides of economically profitable. Namely, it has a triple bottom line: balance between social, environmental and financial profitability, the main characteristic defining ethical banking. Moreover, it exhibits a total transparency in its loan portfolio [37]. The human resources policy of Triodos leads to a sensible lower difference in salaries, where the range of wages is 1 to 9, whilst in conventional banks it ranges from 1 to 600 [10]. It is striking that the capital of Triodos is not listed on the Stock Exchange, so that its value is not exposed to the market volatility and so remains stable.

We have chosen Triodos Bank as a model of ethical banking for several reasons. First, Triodos was a founder of the GABV. Second, it is the most widespread ethical financial institution throughout Europe, since it operates in five European countries: Holland, Spain, Belgium, Germany and United Kingdom. Third, it is in the top of the Social and Ethical Banking Index and, therefore, it holds the higher commitment with ethical banking [15]. Fourth, it is the first of 72 British financial institutions by its social and environmental initiatives [10]. Fifth, it is an indicative of how the social economy financing might develop in the future [37].

The high volatility and uncertainty generated in the international markets during the last financial crisis was caused, mainly, by the liquidity and solvency problems that a good number of financial institutions were suffering, arising the more strict regulation established by Basel III [2]. For this reason, the main aim of this work is to study these parameters in ethical banking, in order to check its strength to face future problems and to make a comparison with conventional banking that demonstrates its reliability to depositors.

To do this, we compare the financial statement of Triodos Bank with two of the main conventional banks of each of the five countries in Europe in which it operates, in the period from 2015 to 2018. Namely, we compare Triodos Bank with ten European conventional banks. Furthermore, we test several important variables that determine the stability, liquidity and solvency of a bank, as well as certain growth indicators to contrast their figures and, therefore, the situation between these ten conventional banks and the thirteen European ethical banks shown in Table 2.

Based on the former arguments, we propose the following research hypotheses:

**Table 2. Key figures of European banks included in the Global Alliance for Banking on values.**

| Bank | Year | Co-workers | Clients number | Loans (USD M) | Client funding (USD M) | Total assets (USD M) | Equity (USD M) | Total revenue (USD M) | Net income (USD M) | ROA | ROE | TIER 1 |
|---|---|---|---|---|---|---|---|---|---|---|---|---|
| Cooperative Bank of Karditsa | 2018 | 36 | 17,267 | 61 | 104 | 120 | 15 | 4.2 | 1.1 | 1.00 | 7.35 | 21.0 |
| | 2017 | 34 | 16,745 | 59 | 97 | 115 | 16 | 3.3 | 0.7 | 0.63 | 4.96 | 18.8 |
| | 2016 | 31 | 14,936 | 50 | 78 | 92 | 13 | 3.3 | 0.3 | 0.36 | 2.60 | 17.9 |
| | 2015 | 31 | 14,454 | 52 | 77 | 91 | 13 | 3.1 | 0.6 | 0.62 | 5.02 | 18.1 |
| Banca Etica | 2018 | 309 | 73,842 | 1,028 | 1,773 | 2,179 | 107 | 53.2 | 3.9 | 0.18 | 3.49 | 12.2 |
| | 2017 | 288 | 65,588 | 976 | 1,644 | 2,069 | 113 | 44.6 | 2.6 | 0.15 | 2.52 | 12.2 |
| | 2016 | 285 | 62,429 | 762 | 1,291 | 1,638 | 90 | 52.5 | 4.8 | 0.31 | 5.22 | 12.5 |
| | 2015 | 269 | 56,442 | 713 | 957 | 1,358 | 87 | 38.6 | 0.8 | 0.06 | 0.97 | 11.3 |
| Alternative Bank Switzerland | 2018 | 111 | 35,588 | 1,396 | 1,629 | 1,830 | 193 | 27.4 | 7.1 | 0.38 | 3.64 | 17.7 |
| | 2017 | 107 | 32,831 | 1,366 | 1,602 | 1,787 | 177 | 27.2 | 6.8 | 0.38 | 3.85 | 16.5 |
| | 2016 | 103 | 21,551 | 1,181 | 1,469 | 1,624 | 146 | 26.4 | 6.9 | 0.41 | 4.57 | 15.1 |
| | 2015 | 96 | 31,616 | 1,066 | 1,477 | 1,602 | 111 | 24.1 | 6.5 | 0.39 | 5.70 | 12.2 |
| Freie Gemeinschaftsbank Genossenschaff | 2018 | 27 | 4,904 | 238 | 268 | 289 | 20 | 4.5 | 0.2 | 0.06 | 0.91 | 11.7 |
| | 2017 | 23 | 4,974 | 222 | 261 | 282 | 20 | 4.1 | 0.1 | 0.04 | 0.51 | 12.1 |
| | 2016 | 19 | 4,843 | 194 | 247 | 267 | 19 | 4.0 | 0.2 | 0.08 | 1.06 | 12.8 |
| | 2015 | 22 | 4,742 | 184 | 248 | 268 | 19 | 3.9 | 0.5 | 0 | 0.05 | 14.0 |
| UnweltBank | 2018 | 163 | 113,920 | 2,869 | 2,803 | 4,235 | 265 | 67.7 | 29.9 | 0.68 | 11.42 | 10.7 |
| | 2017 | 149 | 113,694 | 2,871 | 2,691 | 4,175 | 254 | 66.4 | 31.2 | 0.79 | 13.71 | 10.4 |
| | 2016 | 139 | 114,661 | 2,504 | 2,252 | 3,374 | 201 | 72.1 | 35.6 | 1.00 | 17.94 | 9.9 |
| | 2015 | 132 | 114,681 | 2,639 | 2,183 | 3,008 | 182 | 68.9 | 37.8 | 1.24 | 22.19 | 8.7 |
| GLS Bank | 2018 | 582 | 218,170 | 5,451 | 5,343 | 6,504 | 538 | 128.0 | 10.7 | 0.16 | 8.10 | 12.3 |
| | 2017 | 514 | 212,482 | 5,150 | 4,953 | 6,058 | 485.5 | 115.3 | 8.6 | 0.15 | 9.20 | 11.2 |
| | 2016 | 524 | 210,894 | 4,136 | 4,109 | 4,838 | 356.8 | 95.1 | 6.8 | 0.10 | 8.50 | 10.6 |
| | 2015 | 527 | 193,314 | 3,901 | 3,946 | 4,554 | 325.8 | 91.8 | 5.7 | 0.10 | 8.90 | 11.3 |
| Triodos Bank | 2018 | 1,427 | 715,000 | 8,327 | 10,942 | 12,443 | 1,295 | 314.1 | 45.6 | 0.36 | 3.60 | 17.7 |
| | 2017 | 1,377 | 681,000 | 7,905 | 10,449 | 11,863 | 1,213 | 270.9 | 42.2 | 0.40 | 3.90 | 19.2 |
| | 2016 | 1,271 | 652,000 | 6,008 | 8,446 | 9,558 | 951 | 240.8 | 32.4 | 0.30 | 3.50 | 19.2 |
| | 2015 | 1,121 | 607,000 | 5,689 | 7,943 | 8,956 | 851 | 234.8 | 45.2 | 0.50 | 5.50 | 19.0 |
| Crédit Coopératif | 2018 | 1,931 | 443,889 | 20,025 | 14,180 | 23,674 | 1,857 | 466.4 | 44.3 | 0.18 | 2.31 | 12.0 |
| | 2017 | 1,960 | 342,211 | 19,116 | 14,094 | 22,081 | 1,939 | 484.7 | 60.5 | 0.29 | 3.32 | 12.8 |
| | 2016 | 1,967 | 335,658 | 14,901 | 11,333 | 19,399 | 1,576 | 469.3 | 44.4 | 0.23 | 2.68 | 12.1 |
| | 2015 | 1,982 | 337,100 | 14,424 | 11,145 | 17,377 | 1,528 | 450.8 | 43.5 | 0.25 | 2.83 | 10.7 |
| Ecology Building Society | 2018 | 31 | 10,153 | 161 | 211 | 226 | 13 | 5.3 | 1.4 | 0.57 | 10.11 | 17.5 |
| | 2017 | 29 | 9,989 | 154 | 226 | 241 | 13 | 4.7 | 1.2 | 0.52 | 9.54 | 18.1 |
| | 2016 | 28 | 10,168 | 146 | 201 | 214 | 11 | 5.1 | 1.2 | 0.58 | 10.60 | 17.0 |
| | 2015 | 25 | 9,932 | 168 | 199 | 216 | 11 | 5.2 | 1.3 | 0.62 | 11.35 | 15.1 |
| Opportunity Bank Serbia | 2018 | 378 | 72,000 | 126 | 90 | 167 | 32 | 19.3 | 4.0 | 2.74 | 14.29 | 17.8 |
| | 2017 | 338 | 63,000 | 113 | 72 | 146 | 28 | 16.9 | 3.3 | 2.57 | 13.80 | 16.1 |
| | 2016 | 312 | 54,083 | 86 | 63 | 113 | 21 | 14.4 | 3.0 | 2.69 | 15.14 | 16.8 |
| | 2015 | 284 | 49,516 | 78 | 61 | 106 | 18 | 12.7 | 2.3 | 2.20 | 13.52 | 15.8 |
| Mercur Cooperative Bank | 2018 | 99 | 34,367 | 258 | 479 | 554 | 48 | 21.6 | 0.5 | 0.08 | 0.97 | 16.0 |
| | 2017 | 97 | 29,888 | 282 | 493 | 569 | 51 | 19.0 | -2.9 | -0.50 | -5.90 | 14.5 |
| | 2016 | 94 | 28,464 | 237 | 379 | 446 | 45 | 20.8 | 3.0 | 0.60 | 7.10 | 15.4 |
| | 2015 | 83 | 26,278 | 209 | 336 | 393 | 36 | 16.7 | 1.2 | 0.30 | 3.30 | 14.1 |

(*Continued*)

**Table 2.** (Continued)

| Bank | Year | Co-workers | Clients number | Loans (USD M) | Client funding (USD M) | Total assets (USD M) | Equity (USD M) | Total revenue (USD M) | Net income (USD M) | ROA | ROE | TIER 1 |
|---|---|---|---|---|---|---|---|---|---|---|---|---|
| Ekobanken | 2018 | 19 | 6,109 | 91 | 98 | 108 | 8 | 2.6 | 0.1 | 0.13 | 1.72 | 18.4 |
| | 2017 | 18 | 5,747 | 92 | 96 | 107 | 8 | 2.4 | 0.1 | 0.14 | 1.73 | 22.3 |
| | 2016 | 19 | 5,360 | 70 | 76 | 85 | 7 | 2.0 | -1.0 | -1.13 | -13.34 | 18.6 |
| | 2015 | 19 | 5,000 | 61 | 77 | 87 | 8 | 1.9 | 0.1 | 0.08 | 1.01 | 18.9 |
| Cultura Bank | 2018 | 16 | 6,535 | 73 | 110 | 123 | 10 | 3.5 | 0.7 | 0.55 | 6.20 | 19.3 |
| | 2017 | 18 | 6,003 | 69 | 102 | 115 | 11 | 3.2 | 0.4 | 0.34 | 3.75 | 19.9 |
| | 2016 | 18 | 5,983 | 59 | 86 | 97 | 9 | 3.0 | 0.7 | 0.69 | 7.86 | 18.3 |
| | 2015 | 17 | 5,712 | 52 | 76 | 86 | 7 | 3.0 | 0.2 | 0.22 | 2.51 | 16.0 |
| Total | 2018 | 5,129 | 1,751,114 | 40,104 | 38,030 | 52,452 | 4,401 | 1,117.8 | 149.5 | | | |
| | 2017 | 4,952 | 1,584,152 | 38,375 | 36,780 | 49,608 | 4,328.5 | 1,062.7 | 154.8 | | | |
| | 2016 | 4,810 | 1,531,030 | 30,334 | 30,030 | 41,745 | 3,445.8 | 1,008.8 | 138.3 | | | |
| | 2015 | 4,608 | 1,455,787 | 29,236 | 28,725 | 38,102 | 3,196.8 | 955.5 | 145.7 | | | |
| | % var. | 11.31 | 20.29 | 37.17 | 32.39 | 37.66 | 37.67 | 16.99 | 2.61 | | | |

Source: Compiled by the authors based on www.gabv.org

Hypothesis 1 (H$_1$): Ethical banking has more economic solvency than conventional banking.

Hypothesis 2 (H$_2$): Ethical banking is more linked to the real economy than conventional banking.

Hypothesis 3 (H3): Ethical banking is growing more than conventional banking.

The rest of this article is organized as follows. Section 2 provides a description of the sample, the variables and the methodology to be used. Section 3 presents the results of our comparative analysis between ethical and conventional banking. Section 4 discusses the results and provides the main conclusions derived from this manuscript.

## Materials and methods

In a first stage, the financial data of Triodos Bank, the most important European ethical bank belonging to the GBVA, are compared to two of most important conventional financial companies of each of the five European countries in which Triodos operates, during the period 2015 to 2018. Thus, Triodos Bank is compared to Banco Santander and Banco Bilbao Vizcaya (BBVA), in Spain; ING Bank and Rabobank, in Holland; Deutsche Bank and Commerzbank, in Germany; HSBC Holdings and Lloyds Banking, in UK; and, Dexia and Belfius, in Belgium. By considering that European ethical banks are relatively new banks (which implies that they have experienced their greatest growth in recent years) and since our aim is to analyse the current situation and to compare the relative figures between conventional and ethical banking, the elected period was the past four years.

The first methodology to be applied has consisted in a comparative financial and economic analysis of cases, based on their balance sheet and statement of income [42, 43]. The data have been obtained from the annual reports published in the corporative webs of the involved companies (see "Conventional" sheet in S1 File). A preliminary analysis has been implemented, by comparing the number of employees, the profit per employee and the volume of loans and deposits. Then, by analyzing the balance sheets, we have compared the variation and

composition of the assets (non-current, current and total assets), the liabilities (non-current and current liabilities) and the equity.

Next, we have compared and analyzed the evolution of the main ratios. First, the guarantee ratio [44, 45], which is calculated by dividing total assets by total liabilities, indicates the ability of the bank to cover all its debt obligations with its assets. Second, the liquidity ratio [46, 47], which is calculated by dividing current assets by current liabilities, shows the ability of the bank to meet its short-term obligations; namely, if this ratio is greater than one, the bank will not have liquidity problems in the short-term. Third, indebtedness ratios [48–51], long-term, short-term and total indebtedness, which are calculated by dividing non-current liabilities, current liabilities and total liabilities, respectively, by the equity, indicate the proportion of debt over equity and, in general, they are better the smaller they are.

Afterwards, by analyzing the statement of income, we have compared the variation of income, expenses and net result. Finally, we have calculated and compared the return on assets (ROA) and the return on equity (ROE), in order to study the profitability. Likewise, we have compared the Tier 1 capital, the key measure of the financial strength of a bank, since it is regulated with the aim of assess its solvency; moreover, it evaluates the degree of bank capitalization relative to its assets and by considering the risk generated by the bank activity. According to Basel II, it should be at least 6%, but an optimal level should be equal to 8%.

In a second stage, we have applied a statistical test to the thirteen European ethical banks, whose data have been obtained from the web page of the GABV, and the website of the ten aforementioned conventional banks (see "Ethical" sheet in S1 File). Specifically, we have applied a means difference test and an analysis of variance (ANOVA) on the main ratios (total indebtedness, guarantee, coverage, loans to assets and funding to assets) and growth indicators (loans, funding, assets and income), in order to check if, in effect, the differences previously observed are significant.

A distinctive feature of ethical versus conventional banking is that, as seen in the previous section, ethical banks focus their operations on real economy, by neglecting speculative operations. However, publicly available financial information does not provide this differentiation between bank activities. For this reason, we will use lending and deposit information as a proxy for distinguish between the real and financial economy [8].

Finally, we have considered a dummy variable, which takes the value 1 if the bank is ethical and 0 if it is conventional. To check the robustness of the obtained results, we have performed probit regressions between each of the analyzed variables in the previous statistical test and the dummy variable. In all of them, ROE ratio has been considered as a control variable.

## Results

### Preliminary analysis

According to Table 3, between 2015 and 2018, the behavior experienced by the number of employees in Triodos Bank has been very different from that of the conventional banks, since Triodos has had an increase of 27.30%, whilst the rest of banks have diminished, except Banco Santander that have increased by around 4.57%. The decrease has reached the 35.74% in Dexia, the 19.52% in Rabobank and the 10.90% in HSBC Holdings.

The volume of loans and deposits showed a similar situation. The volume of loans increased by 39.46% in Triodos and the volume of deposits by 31.25% between 2015 and 2018. Meanwhile, the conventional banks have decreased or have experienced far lower growths. However, with respect to the profit per employee, the situation is not clear, since there is not an identifiable pattern over the analyzed period.

**Table 3. Comparison of the number of employees, profit per employee, volume of loans and deposits.**

| Bank | Year | Number of employees | % var. | Profit per employe (€ M) | % var. | Volume of loans (€ M) | %var. | Volume of deposits (€ M) | %var. |
|---|---|---|---|---|---|---|---|---|---|
| Triodos Bank | 2018 | 1,427 | 27.30 | 0.027 | -25.54 | 7,274 | 39.46 | 9,558 | 31.25 |
| | 2017 | 1,377 | | 0.027 | | 6,598 | | 8,722 | |
| | 2016 | 1,271 | | 0.023 | | 5,708 | | 8,025 | |
| | 2015 | 1,121 | | 0.036 | | 5,216 | | 7,283 | |
| Banco Santander | 2018 | 202,713 | 4.57 | 0.046 | 21.47 | 882,921 | 11.64 | 780,496 | 14.25 |
| | 2017 | 202,251 | | 0.044 | | 848,914 | | 777,730 | |
| | 2016 | 188,492 | | 0.042 | | 790,470 | | 691,112 | |
| | 2015 | 193,863 | | 0.040 | | 790,848 | | 683,142 | |
| BBVA | 2018 | 125,627 | -8.94 | 0.049 | 102.98 | 419,660 | -11,06 | 435,229 | -14.99 |
| | 2017 | 131,856 | | 0.036 | | 445,275 | | 467,949 | |
| | 2016 | 134,792 | | 0.035 | | 483,672 | | 499,706 | |
| | 2015 | 137,968 | | 0.024 | | 471,828 | | 511,992 | |
| ING Bank | 2018 | 52,233 | -0.92 | 0.092 | 2.64 | 592,196 | -15.40 | 555,812 | -17,31 |
| | 2017 | 51,504 | | 0.099 | | 574,899 | | 552,690 | |
| | 2016 | 51,943 | | 0.083 | | 562,873 | | 531,096 | |
| | 2015 | 52,720 | | 0.090 | | 700,007 | | 672,204 | |
| Rabobank | 2018 | 41,861 | -19.52 | 0.072 | 68.59 | 443,867 | -4.75 | 342,410 | -1.01 |
| | 2017 | 43,810 | | 0.061 | | 432,564 | | 340,682 | |
| | 2016 | 45,567 | | 0.044 | | 452,807 | | 347,712 | |
| | 2015 | 52,013 | | 0.043 | | 466,000 | | 345,900 | |
| Deutsche Bank | 2018 | 91,737 | -9.26 | 0.004 | 105.55 | 400,297 | -6.42 | 564,405 | -0.45 |
| | 2017 | 97,535 | | -0.008 | | 401,699 | | 580,812 | |
| | 2016 | 99,744 | | -0.014 | | 408,909 | | 550,204 | |
| | 2015 | 101,104 | | -0.067 | | 427,749 | | 566,974 | |
| Commerzbank | 2018 | 43,412 | -4.42 | 0.022 | -15.53 | 279,137 | -3.97 | 346,668 | -9.94 |
| | 2017 | 43,560 | | 0.006 | | 265,712 | | 341,260 | |
| | 2016 | 44,267 | | 0.009 | | 276,578 | | 356,362 | |
| | 2015 | 45,419 | | 0.026 | | 290,680 | | 384,938 | |
| HSBC Holdings | 2018 | 235,217 | -10,90 | 0.064 | 11.71 | 981,696 | 6.19 | 1,362,643 | 5.67 |
| | 2017 | 229,000 | | 0.052 | | 962,964 | | 1,364,462 | |
| | 2016 | 241,000 | | 0.014 | | 861,504 | | 1,272,386 | |
| | 2015 | 264,000 | | 0.057 | | 924,454 | | 1,289,586 | |
| Lloyds Banking | 2018 | 80,117 | -10.48 | 0.055 | 414.16 | 444,400 | -2.37 | 416,000 | -0.56 |
| | 2017 | 81,667 | | 0.043 | | 455,700 | | 418,124 | |
| | 2016 | 86,516 | | 0.029 | | 457,958 | | 415,460 | |
| | 2015 | 89,501 | | 0.011 | | 455,175 | | 418,326 | |
| Dexia | 2018 | 773 | -35.74 | -0.646 | -559.52 | 35,158 | -72.51 | 4,873 | -48.15 |
| | 2017 | 996 | | -0.463 | | 99,264 | | 6,404 | |
| | 2016 | 1,148 | | 0.307 | | 119,206 | | 10,778 | |
| | 2015 | 1,203 | | 0.140 | | 127,876 | | 9,399 | |
| Belfius | 2018 | 6,494 | -1.62 | 0.100 | 30.58 | 91,123 | 4.51 | 79,661 | 16.87 |
| | 2017 | 6,432 | | 0.094 | | 90,057 | | 76,274 | |
| | 2016 | 6,429 | | 0.083 | | 89,702 | | 74,171 | |
| | 2015 | 6,601 | | 0.077 | | 87,189 | | 68,163 | |

Source: Compiled by the authors based on the bank's annual reports.

## Balance sheet

Table 4 shows the evolution of assets. We find again a great difference in the volume of total assets between Triodos and conventional banks, since Triodos has increased its total assets during the period from 2015 to 2018 by 32.38%. However, eight of the conventional banks have diminished their investment and only two banks have increased, but only by 6.16% (HSBC Holdings) and 8.88% (Banco Santander).

With respect to the distribution between current and non-current assets, we can observe that Triodos is between the three banks whose current assets exceed 96%, namely Triodos has a high commitment with the short-term investment [43]. In fact, this entity shows the largest increase in current assets, by 32.56%. The next increases have been the ones experienced by Banco Santander, by 9.34%, and HSBC Holding, by 8.31%; observe that both of them are far from Triodos.

Therefore, Triodos invests a low percentage in non-current assets, which is in line with the few number of its branches, by considering that the higher proportion of its business is based on the internet, as explained in Section 1 of this manuscript.

As shown by Table 5, the coverage ratio (proportion of equity over total assets) is the highest in Triodos, greater than 10%. The rest of the banks have less than 8% of equity. With respect to the current liabilities, most banks are above 80%, except Lloyds Banking and Belfius, which are around 64%. Moreover, Triodos has the lowest non-current liabilities ratio between total assets, by exceeding just 1%. The other banks also have low percentages, below 16%, except for, again, Lloys Banking and Belfius, which present values around 30%.

Table 6 shows the guarantee, liquidity and indebtedness ratios. Guarantee ratio is in all banks greater than 1, which indicates that banking have enough assets to cover its debt. It is important to highlight that Triodos has the highest ratio. With respect to the liquidity ratio, Triodos is in the top, along with ING Bank, Deutsche Bank and Lloyds Banking, which indicates that it has no problems in the short-term to attend its debt obligations.

According to the highest equity ratio, the total indebtedness is the lowest in Triodos, since it is around 9% and being far below the rest of banks. Analogously, Triodos presents the lowest short- and long-term indebtedness ratios. Not only has Triodos the best figures in all years, but it has also experienced a decrease during the considered period. Specifically, Triodos reduced its long-term indebtedness ratio by 27.71%, its short-term indebtedness ratio by 9.30% and its total indebtedness ratio by 9.57%. This supposes a solvency, in general, better than that of its competitors.

## Statement of income

Table 7 includes income, expenses and net result, three variables contained in the statement of income. Moreover, it shows two well-known accounting measures of profitability, ROA and ROE. Finally, we have considered interesting to finish the financial and economic analysis, by comparing the Tier 1, since it is a compulsory measure of banks, which analyzes its financial strength.

In the analyzed period, seven of the ten conventional banks reduced their income, whilst Triodos experienced an increase of 25.82%, which is higher than 6.16% of Lloyds Banking, 8.10% of Belfius and 8.12% of BBVA, the three banks that have increased. With respect to the expenses, conventional banks managed to reduce them; however, Triodos increased its expenses by 20.55%. The main reason for this is the increasing number of employees, since in this period workers increased by 27.30%, as seen above, which is a logic consequence of its expansion. Therefore, since expenses increased more than income in Triodos, it is logical that the net result experienced a reduction of 5.22%. However, three of the conventional banks

**Table 4. Comparison of assets.**

| Bank | Year | Non-current assets (€ M) | % | %var. | Current assets (€ M) | % | %var. | Total assets (€ M) | %var. |
|---|---|---|---|---|---|---|---|---|---|
| Triodos Bank | 2018 | 332 | 3.05 | 26.94 | 10,538 | 96.95 | 32.56 | 10,870 | 32.38 |
| | 2017 | 295 | 2.98 | | 9,608 | 97.02 | | 9,902 | |
| | 2016 | 275 | 3.03 | | 8,806 | 96.97 | | 9,081 | |
| | 2015 | 261 | 3.18 | | 7,950 | 96.82 | | 8,211 | |
| Banco Santander | 2018 | 117,349 | 8.04 | 3.91 | 1,341,922 | 91.96 | 9.34 | 1,459,271 | 8.88 |
| | 2017 | 136,787 | 9.47 | | 1,307,518 | 90.53 | | 1,444,305 | |
| | 2016 | 126,097 | 9.42 | | 1,213,028 | 90.58 | | 1,339,125 | |
| | 2015 | 112,928 | 8.43 | | 1,227,332 | 91.57 | | 1,340,260 | |
| BBVA | | 45,931 | 6.79 | -16.00 | 630,758 | 93.21 | -9.27 | 676,689 | -9.76 |
| | 2017 | 78,978 | 11.45 | | 611,081 | 88.55 | | 690,059 | |
| | 2016 | 69,607 | 9.51 | | 662,249 | 90.49 | | 731,856 | |
| | 2015 | 54,682 | 7.29 | | 695,173 | 92.71 | | 749,855 | |
| ING Bank | 2018 | 15,440 | 1.74 | -13.26 | 871,590 | 98,26 | -11.44 | 887,030 | -11.47 |
| | 2017 | 18,421 | 2.18 | | 827,897 | 97.82 | | 846,318 | |
| | 2016 | 20,447 | 2.42 | | 823,472 | 97.58 | | 843,919 | |
| | 2015 | 17,800 | 1.78 | | 984,192 | 98.22 | | 1,001,992 | |
| Rabobank | 2018 | 16,095 | 2.73 | -32.94 | 574,342 | 97.27 | -12.29 | 590,437 | -13.02 |
| | 2017 | 19,164 | 3.18 | | 583,827 | 96.82 | | 602,991 | |
| | 2016 | 19,079 | 2.88 | | 643,514 | 97.12 | | 662,593 | |
| | 2015 | 24,000 | 3.54 | | 654,800 | 96.46 | | 678,800 | |
| Deutsche Bank | 2018 | 113,206 | 8.40 | -47.27 | 1,234,931 | 91.60 | -12.69 | 1,348,137 | -17.25 |
| | 2017 | 124,177 | 8.42 | | 1,350,555 | 91.58 | | 1,474,732 | |
| | 2016 | 151,262 | 9.51 | | 1,439,284 | 90.49 | | 1,590,546 | |
| | 2015 | 214,704 | 13.18 | | 1,414,426 | 86.82 | | 1,629,130 | |
| Commerzbank | 2018 | 26,085 | 5.64 | 94.17 | 436,284 | 94.36 | -15.98 | 462,369 | -13.20 |
| | 2017 | 12,489 | 2.76 | | 440,011 | 97.24 | | 452,500 | |
| | 2016 | 21,148 | 4.40 | | 459,252 | 95.60 | | 480,400 | |
| | 2015 | 13,434 | 2.52 | | 519,267 | 97.48 | | 532,701 | |
| HSBC Holdings | 2018 | 611,548 | 23.91 | -0.15 | 1,946,575 | 76.09 | 8.31 | 2,558,124 | 6.16 |
| | 2017 | 548,960 | 21.77 | | 1,972,811 | 78.23 | | 2,521,771 | |
| | 2016 | 585,620 | 24.66 | | 1,789,366 | 75.34 | | 2,374,986 | |
| | 2015 | 612,447 | 25.42 | | 1,797,209 | 74.58 | | 2,409,656 | |
| Lloyds Banking | 2018 | 64,432 | 8.08 | -21.42 | 733,166 | 91.92 | 1.17 | 797,598 | -1.13 |
| | 2017 | 82,124 | 10.11 | | 729,985 | 89.89 | | 812,109 | |
| | 2016 | 94,772 | 11.59 | | 723,021 | 88.41 | | 817,793 | |
| | 2015 | 82,000 | 10.17 | | 724,688 | 89.83 | | 806,688 | |
| Dexia | 2018 | 24,862 | 15.66 | -36.18 | 133,942 | 84.34 | -29.99 | 158,804 | -31.04 |
| | 2017 | 34,492 | 19.06 | | 146,446 | 80.94 | | 180,938 | |
| | 2016 | 38,910 | 18.29 | | 173,861 | 81.71 | | 212,771 | |
| | 2015 | 38,954 | 16.92 | | 191,327 | 83.08 | | 230,281 | |
| Belfius | 2018 | 51,621 | 31.44 | -20.43 | 112,544 | 68.56 | 0.41 | 164,165 | -7.23 |
| | 2017 | 53,544 | 31.88 | | 114,415 | 68.12 | | 167,959 | |
| | 2016 | 59,905 | 33.90 | | 116,816 | 66.10 | | 176,721 | |
| | 2015 | 64,879 | 36.66 | | 112,083 | 63.34 | | 176,962 | |

Source: Compiled by the authors based on the bank's annual reports.

**Table 5. Comparison of equity and liabilities.**

| Bank | Year | Non-current liabilites (€ M) | % | %var. | Current liabilities (€ M) | % | %var. | Equity (€ M) | % | %var. |
|---|---|---|---|---|---|---|---|---|---|---|
| Triodos Bank | 2018 | 113 | 1.04 | 4.78 | 9,625 | 88.55 | 31.45 | 1,131 | 10.41 | 44.93 |
| | 2017 | 103 | 1.04 | | 8,786 | 88.73 | | 1,013 | 10.23 | |
| | 2016 | 121 | 1.33 | | 8,056 | 88.71 | | 904 | 9.95 | |
| | 2015 | 108 | 1.32 | | 7,322 | 89.18 | | 781 | 9.51 | |
| Banco Santander | 2018 | 41.879 | 2.87 | -0.71 | 1,310,031 | 89.77 | 9.23 | 107,361 | 7.36 | 8.72 |
| | 2017 | 44,163 | 3.06 | | 1,293,310 | 89.55 | | 106,832 | 7.40 | |
| | 2016 | 43,158 | 3.22 | | 1,193,268 | 89.11 | | 102,699 | 7.67 | |
| | 2015 | 42,178 | 3.15 | | 1,199,329 | 89.48 | | 98,753 | 7.37 | |
| BBVA | 2018 | 14,349 | 2.12 | -20.80 | 609,466 | 90.07 | -9.90 | 52,874 | 7.81 | -4.36 |
| | 2017 | 15,326 | 2.22 | | 621,410 | 90.05 | | 53,323 | 7.73 | |
| | 2016 | 18,718 | 2.56 | | 657,710 | 89.87 | | 55,428 | 7.57 | |
| | 2015 | 18,118 | 2.42 | | 676,455 | 90.21 | | 55,282 | 7.37 | |
| ING Bank | 2018 | 148,638 | 16.76 | 0.36 | 686,657 | 77.41 | -15.48 | 51,735 | 5.83 | 24.68 |
| | 2017 | 124,499 | 14.71 | | 677,442 | 80.05 | | 44,377 | 5.24 | |
| | 2016 | 137,149 | 16.25 | | 662,624 | 78.52 | | 44,146 | 5.23 | |
| | 2015 | 148,108 | 14.78 | | 812,389 | 81.08 | | 41,495 | 4.14 | |
| Rabobank | 2018 | 18,305 | 3.10 | 7.68 | 529.896 | 89.75 | -14.62 | 42,236 | 7.15 | 2.51 |
| | 2017 | 18,607 | 3.09 | | 544,774 | 90.35 | | 39,610 | 6.57 | |
| | 2016 | 19,294 | 2.91 | | 602,775 | 90.97 | | 40,524 | 6.12 | |
| | 2015 | 17,000 | 2.50 | | 620,600 | 91.43 | | 41,200 | 6.07 | |
| Deutsche Bank | 2018 | 159,418 | 11.83 | -0.37 | 1.119.982 | 83.08 | -20.09 | 68,737 | 5.10 | 1.64 |
| | 2017 | 171,772 | 11.65 | | 1,234,861 | 83.73 | | 68,099 | 4.62 | |
| | 2016 | 191,477 | 12.04 | | 1,334,250 | 83.89 | | 64,819 | 4.08 | |
| | 2015 | 160,016 | 9.82 | | 1,401,489 | 86.03 | | 67,625 | 4.15 | |
| Commerzbank | 2018 | 15,790 | 3.42 | 46.43 | 417.168 | 90.22 | -15.13 | 29,411 | 6.36 | -3.25 |
| | 2017 | 7,022 | 1.55 | | 415,478 | 91.82 | | 30,000 | 6.63 | |
| | 2016 | 7,127 | 1.48 | | 443,673 | 92.35 | | 29,600 | 6.16 | |
| | 2015 | 10,783 | 2.02 | | 491,518 | 92.27 | | 30,400 | 5.71 | |
| HSBC Holdings | 2018 | 340,246 | 13.30 | 13.91 | 2,023,629 | 79.11 | 5.76 | 194,249 | 7.59 | -1.66 |
| | 2017 | 263,903 | 10.46 | | 2,059,997 | 81.69 | | 197,871 | 7.85 | |
| | 2016 | 250,783 | 10.56 | | 1,941,625 | 81.75 | | 182,578 | 7.69 | |
| | 2015 | 298,688 | 12.40 | | 1,913,450 | 79.41 | | 197,518 | 8.20 | |
| Lloyds Banking | 2018 | 245,353 | 30.76 | 3.09 | 502,046 | 62.94 | -3.84 | 50,199 | 6.29 | 7.75 |
| | 2017 | 238,037 | 29.31 | | 524,929 | 64.64 | | 49,143 | 6.05 | |
| | 2016 | 247,706 | 30.29 | | 521,272 | 63.74 | | 48,815 | 5.97 | |
| | 2015 | 238,000 | 29.50 | | 522,099 | 64.72 | | 46,589 | 5.78 | |
| Dexia | 2018 | 24,960 | 15.72 | 248.51 | 126,003 | 79.34 | -42.35 | 7,841 | 4.94 | 72.44 |
| | 2017 | 6,369 | 3.52 | -11.07 | 169,167 | 93.49 | | 5,402 | 2.99 | |
| | 2016 | 5,388 | 2.53 | | 202,809 | 95.32 | | 4,574 | 2.15 | |
| | 2015 | 7,162 | 3.11 | | 218,572 | 94.92 | | 4,547 | 1.97 | |
| Belfius | 2018 | 47,265 | 28.79 | -19.55 | 106,940 | 65.14 | -2.38 | 9,960 | 6.07 | 15.01 |
| | 2017 | 44,490 | 26.49 | -24.27 | 113,948 | 67.84 | | 9,521 | 5.67 | |
| | 2016 | 54,953 | 31.10 | | 112,756 | 63.80 | | 9,012 | 5.10 | |
| | 2015 | 58,751 | 33.20 | | 109,551 | 61.91 | | 8,660 | 4.89 | |

Source: Compiled by the authors based on the bank's annual reports.

**Table 6. Guarantee, coverage, liquidity and indebtedness ratios.**

| Bank | Year | Guarantee ratio | % var. | Coverage ratio | % var. | Liquidity ratio | % var. | Long-term indebtedness | % var. | Short-term indebdedness | % var. | Total indebtedness | % var. |
|---|---|---|---|---|---|---|---|---|---|---|---|---|---|
| Triodos Bank | 2018 | 1.116 | 1.01 | 0.104 | 9.48 | 1.095 | 0.84 | 0.100 | -27.71 | 8.509 | -9.30 | 8.609 | -9.57 |
| | 2017 | 1.114 | | 0.102 | | 1.093 | | 0.102 | | 8.670 | | 8.776 | |
| | 2016 | 1.114 | | 0.100 | | 1.093 | | 0.134 | | 8.915 | | 9.049 | |
| | 2015 | 1.105 | | 0.095 | | 1.086 | | 0.139 | | 9.381 | | 9.520 | |
| Banco Santander | 2018 | 1.079 | -0.01 | 0.074 | -0.15 | 1.024 | 0.10 | 0.390 | -8.67 | 12.202 | 0.47 | 12.592 | 0.16 |
| | 2017 | 1.080 | | 0.074 | | 1.011 | | 0.413 | | 12.106 | | 12.519 | |
| | 2016 | 1.083 | | 0.077 | | 1.017 | | 0.420 | | 11.619 | | 12.039 | |
| | 2015 | 1.080 | | 0.074 | | 1.023 | | 0.427 | | 12.145 | | 12.572 | |
| BBVA | 2018 | 1.085 | 0.48 | 0.078 | 5.99 | 1.035 | 0.71 | 0.271 | -17.20 | 11.527 | -5.80 | 11.798 | -6.10 |
| | 2017 | 1.084 | | 0.077 | | 0.983 | | 0.287 | | 11.654 | | 11.941 | |
| | 2016 | 1.082 | | 0.076 | | 1.007 | | 0.338 | | 11.866 | | 12.204 | |
| | 2015 | 1.080 | | 0.074 | | 1.028 | | 0.328 | | 12.236 | | 12.564 | |
| ING Bank | 2018 | 1.062 | 1.80 | 0.058 | 40.84 | 1.269 | 4.77 | 2.873 | -19.51 | 13.273 | -32.21 | 16.146 | -30.25 |
| | 2017 | 1.055 | | 0.052 | | 1.222 | | 2.805 | | 15.266 | | 18.071 | |
| | 2016 | 1.055 | | 0.052 | | 1.243 | | 3.107 | | 15.010 | | 18.117 | |
| | 2015 | 1.043 | | 0.041 | | 1.211 | | 3.569 | | 19.578 | | 23.147 | |
| Rabobank | 2018 | 1.077 | 1.17 | 0.072 | 17.86 | 1.084 | 2.73 | 0.433 | 5.04 | 12.546 | -16.71 | 12.979 | -16.13 |
| | 2017 | 1.070 | | 0.066 | | 1.072 | | 0.470 | | 13.753 | | 14.223 | |
| | 2016 | 1.065 | | 0.061 | | 1.068 | | 0.476 | | 14.875 | | 15.351 | |
| | 2015 | 1.065 | | 0.061 | | 1.055 | | 0.413 | | 15.063 | | 15.476 | |
| Deutsche Bank | 2018 | 1.054 | 1.00 | 0.051 | 22.83 | 1.103 | 9.25 | 2.319 | -1.99 | 16.294 | -21.38 | 18.613 | -19.39 |
| | 2017 | 1.048 | | 0.046 | | 1.094 | | 2.522 | | 18.133 | | 20.656 | |
| | 2016 | 1.042 | | 0.041 | | 1.079 | | 2.954 | | 20.584 | | 23.538 | |
| | 2015 | 1.043 | | 0.042 | | 1.009 | | 2.366 | | 20.724 | | 23.091 | |
| Commerzbank | 2018 | 1.068 | 0.70 | 0.064 | 11.46 | 1.046 | -1.01 | 0.537 | 51.36 | 14.184 | -12.27 | 14.721 | -10.91 |
| | 2017 | 1.071 | | 0.066 | | 1.059 | | 0.234 | | 13.849 | | 14.083 | |
| | 2016 | 1.066 | | 0.062 | | 1.035 | | 0.241 | | 14.989 | | 15.230 | |
| | 2015 | 1.061 | | 0.057 | | 1.056 | | 0.355 | | 16.168 | | 16.523 | |
| HSBC Holdings | 2018 | 1.082 | -0.65 | 0.076 | -7.36 | 0.962 | 2.41 | 1.752 | 15.83 | 10.418 | 7.54 | 12.169 | 8.66 |
| | 2017 | 1.085 | | 0.078 | | 0.958 | | 1.334 | | 10.411 | | 11.745 | |
| | 2016 | 1.083 | | 0.077 | | 0.922 | | 1.374 | | 10.634 | | 12.008 | |
| | 2015 | 1.089 | | 0.082 | | 0.939 | | 1.512 | | 9.687 | | 11.200 | |
| Lloyds Banking | 2018 | 1.067 | 0.55 | 0.063 | 8.98 | 1.460 | 5.21 | 4.888 | -4.32 | 10.001 | -10.76 | 14.889 | -8.74 |
| | 2017 | 1.064 | | 0.061 | | 1.391 | | 4.844 | | 10.682 | | 15.525 | |
| | 2016 | 1.063 | | 0.060 | | 1.387 | | 5.074 | | 10.679 | | 15.753 | |
| | 2015 | 1.061 | | 0.058 | | 1.388 | | 5.109 | | 11.206 | | 16.315 | |
| Dexia | 2018 | 1.052 | 3.12 | 0.049 | 150.06 | 1.063 | 21.44 | 3.183 | 102.10 | 16.070 | -66.57 | 19.253 | -61.22 |
| | 2017 | 1.031 | | 0.030 | | 0.866 | | 1.179 | | 31.316 | | 32.495 | |
| | 2016 | 1.022 | | 0.021 | | 0.857 | | 1.178 | | 44.340 | | 45.517 | |
| | 2015 | 1.020 | | 0.020 | | 0.875 | | 1.575 | | 48.069 | | 49.645 | |
| Belfius | 2018 | 1.065 | 1.25 | 0.061 | 23.98 | 1.052 | 2.86 | 4.745 | -30.05 | 10.737 | -15.12 | 15.482 | -20.33 |
| | 2017 | 1.060 | | 0.057 | | 1.004 | | 4.673 | | 11.968 | | 16.641 | |
| | 2016 | 1.054 | | 0.051 | | 1.036 | | 6.098 | | 12.512 | | 18.610 | |
| | 2015 | 1.051 | | 0.049 | | 1.023 | | 6.784 | | 12.650 | | 19.434 | |

Source: Compiled by the authors based on the bank's annual reports.

**Table 7. Comparison of income, expenses, taxes and net result by year.**

| Bank | Year | Income (€ M) | % var. | Expenses (€ M) | % var. | Net result (€ M) | %var. | ROA | %var. | ROE | %var. | TIER 1 | %var. |
|---|---|---|---|---|---|---|---|---|---|---|---|---|---|
| Triodos Bank | 2018 | 266 | 25.82 | 216 | 20.55 | 39 | -5.22 | 0.46 | -29.52 | 3.41 | -34.60 | 17.70 | -6.84 |
| | 2017 | 240 | | 191 | | 37 | | 0.50 | | 3.69 | | 19.20 | |
| | 2016 | 218 | | 179 | | 29 | | 0.42 | | 3.24 | | 19.20 | |
| | 2015 | 212 | | 158 | | 41 | | 0.66 | | 5.22 | | 19.00 | |
| Banco Santander | 2018 | 77,542 | -2.19 | 63,341 | -9.16 | 9,315 | 27.01 | 0.97 | 36.62 | 8.68 | 16.83 | 13.12 | 4.54 |
| | 2017 | 78,155 | | 66,064 | | 8,207 | | 0.84 | | 7.68 | | 12.77 | |
| | 2016 | 76,578 | | 65,810 | | 7,486 | | 0.80 | | 7.29 | | 12.53 | |
| | 2015 | 79,278 | | 69,731 | | 7,334 | | 0.71 | | 7.43 | | 12.55 | |
| BBVA | 2018 | 43,145 | 8.12 | 34,699 | -1.71 | 6,151 | 84.83 | 1.25 | 103.33 | 11.63 | 93.24 | 13.20 | 9.09 |
| | 2017 | 43,872 | | 36,941 | | 4,762 | | 1.00 | | 8.93 | | 13.00 | |
| | 2016 | 42,207 | | 35,815 | | 4,693 | | 0.87 | | 8.47 | | 12.90 | |
| | 2015 | 39,904 | | 35,301 | | 3,328 | | 0.61 | | 6.02 | | 12.10 | |
| ING Bank | 2018 | 33,592 | -35.18 | 26,754 | -41.08 | 4,811 | 1.69 | 0.77 | 20.41 | 9.30 | -18.44 | 16.18 | 16.82 |
| | 2017 | 49,232 | | 41,828 | | 5,101 | | 0.87 | | 11.49 | | 16.37 | |
| | 2016 | 49,566 | | 43,629 | | 4,302 | | 0.70 | | 9.74 | | 14.70 | |
| | 2015 | 51,823 | | 45,408 | | 4,731 | | 0.64 | | 11.40 | | 13.85 | |
| Rabobank | 2018 | 11,352 | -5.77 | 7,446 | -18.87 | 3,004 | 35.68 | 0.66 | 56.52 | 7.11 | 32.35 | 19.50 | 18.90 |
| | 2017 | 11,496 | | 7,864 | | 2,674 | | 0.60 | | 6.75 | | 18.80 | |
| | 2016 | 11,622 | | 8,904 | | 2,024 | | 0.41 | | 4.99 | | 17.60 | |
| | 2015 | 12,047 | | 9,178 | | 2,214 | | 0.42 | | 5.37 | | 16.40 | |
| Deutsche Bank | 2018 | 36,917 | -15.35 | 35,587 | -28.41 | 341 | 105.04 | 0.10 | -126.36 | 0.50 | 104.95 | 14.90 | 21.14 |
| | 2017 | 38,162 | | 36,934 | | -735 | | 0.08 | | -1.08 | | 15.40 | |
| | 2016 | 40,944 | | 41,754 | | -1,356 | | -0.05 | | -2.09 | | 13.10 | |
| | 2015 | 43,611 | | 49,708 | | -6,772 | | -0.37 | | -10.01 | | 12.30 | |
| Commerzbank | 2018 | 8,570 | -12.51 | 7,325 | -8.06 | 968 | -19.27 | 0.27 | -21.53 | 3.29 | -16.55 | 13.40 | -2.90 |
| | 2017 | 9,239 | | 8,744 | | 250 | | 0.11 | | 0.83 | | 15.20 | |
| | 2016 | 9,399 | | 8,756 | | 382 | | 0.13 | | 1.29 | | 13.90 | |
| | 2015 | 9,795 | | 7,967 | | 1,199 | | 0.34 | | 3.94 | | 13.80 | |
| HSBC Holdings | 2018 | 66,123 | -10.22 | 46,233 | -15.60 | 15,025 | -0.47 | 0.78 | -0.70 | 7.73 | 1.20 | 16.60 | 19.42 |
| | 2017 | 66,151 | | 48,984 | | 11,879 | | 0.68 | | 6.00 | | 16.40 | |
| | 2016 | 62,190 | | 55,078 | | 3,446 | | 0.30 | | 1.89 | | 16.10 | |
| | 2015 | 73,648 | | 54,781 | | 15,096 | | 0.78 | | 7.64 | | 13.90 | |
| Lloyds Banking | 2018 | 18,724 | 6.16 | 12,764 | -20.19 | 4,400 | 360.25 | 0.75 | 266.66 | 8.77 | 327.15 | 18.20 | 10.98 |
| | 2017 | 18,525 | | 13,250 | | 3,547 | | 0.65 | | 7.22 | | 17.20 | |
| | 2016 | 17,500 | | 13,262 | | 2,514 | | 0.52 | | 5.15 | | 17.00 | |
| | 2015 | 17,637 | | 15,993 | | 956 | | 0.20 | | 2.05 | | 16.40 | |
| Dexia | 2018 | 8,025 | -29.89 | 8,484 | -24.65 | -499 | -395.27 | -0.29 | -455.93 | -6.36 | -271.23 | 26.70 | 67.92 |
| | 2017 | 10,007 | | 10,455 | | -461 | | -0.25 | | -8.53 | | 19.50 | |
| | 2016 | 10,678 | | 10,368 | | 352 | | 0.15 | | 7.70 | | 16.20 | |
| | 2015 | 11,446 | | 11,259 | | 169 | | 0.08 | | 3.72 | | 15.90 | |
| Belfius | 2018 | 2,361 | 8.10 | 1,494 | -0.53 | 650 | 28.46 | 0.53 | 37.04 | 6.53 | 11.69 | 17.00 | 14.09 |
| | 2017 | 2,355 | | 1,392 | | 606 | | 0.57 | | 6.36 | | 15.90 | |
| | 2016 | 2,259 | | 1,479 | | 535 | | 0.44 | | 5.94 | | 16.10 | |
| | 2015 | 2,184 | | 1,502 | | 506 | | 0.39 | | 5.84 | | 14.90 | |

Source: Compiled by the authors based on the bank's annual reports.

suffered important falls in their net results; two of them, Deutsche Bank and Dexia, even had negative figures.

Regarding to ROA and ROE, we do not find a common pattern in conventional banks and different in Triodos. Only Banco Santander, BBVA and ING Bank had a higher ROA than Triodos in the three considered years, and except Deutsche Bank, Commerzbank and Dexia, that have lower ROA than Triodos, the rest of the banks have a similar behavior. Five of the ten conventional banks had higher ROE than Triodos and three had lower ROE than Triodos over all the period; therefore, the data are not conclusive to this respect. However, it is important to remark that, as previously seen, the equity in Triodos is noticeably higher and, consequently, it is expected that the ROE diminished.

Finally, the conventional banks that present greater Tier 1 are Rabobank, Lloyds Bank and Dexia, but Triodos exhibits better values, in general, over the considered period, which is a sign of higher financial power to face creditors.

## Statistical test

In our study, we have observed several differences between Triodos Bank and the conventional analyzed banks. In order to check if these differences are effectively significant between ethical and conventional banks, we have applied the *t* mean difference test and the ANOVA analysis to several meaningful variables, by considering two groups. On the one hand, the ten considered conventional European banks and, on the other hand, the thirteen European ethical banks included in Table 2. We have considered four observations for each bank, namely, the years 2015, 2016, 2017 and 2018. Since we have compared ratios, neither the different size nor the different currencies are considered as a problem. Table 8 shows the obtained results.

Tier 1 is not a variable that defines a kind of banking, since it does not present a significant mean difference. However, ethical banking has less total indebtedness and higher guarantee and coverage ratio than conventional banking, at a 1% significance level, which indicates that ethical banking presents more solvency than conventional banking, as reflected in H1. Similar findings have been obtained in previous studies [8, 14, 52, 53].

The ratio loans to assets and the ratio funding to assets are noticeably higher in ethical banking, specially the second one, at a 1% significance level. This evidences that ethical banking is, in effect, linked to the real economy, by performing an intermediation function between

**Table 8. Comparison between ethical and traditional banks.**

| Variable | *t*-test | | | Anova | | |
|---|---|---|---|---|---|---|
| | Mean traditional banks | Mean ethical banks | Mean difference | F | p-value | Adj. $R^2$ |
| Tier 1 | 15.5393 | 15.1885 | 0.3508 | 0.29 | 0.5937 | – |
| Indebtedness (total) | 17.6219 | 11.4068 | 6.2151 | 23.43** | 0.0000 | 0.1977 |
| Guarantee ratio | 1.0637 | 1.1011 | -0.0374 | 24.29** | 0.0000 | 0.2038 |
| Coverage ratio | 0.0597 | 0.0904 | -0.0307 | 26.43** | 0.0000 | 0.2184 |
| Loans/Assets ratio | 0.5403 | 0.6885 | -0.1482 | 28.50** | 0.0000 | 0.2320 |
| Funding/Assets ratio | 0.5032 | 0.8142 | -0.3110 | 90.44** | 0.0000 | 0.4957 |
| Variation Loans | -9.4123 | 32.2241 | -41.6364 | 23.64** | 0.0001 | 0.5071 |
| Variation Funding | -5.5638 | 33.0085 | -38.5723 | 21.30** | 0.0001 | 0.4799 |
| Variation Assets | -8.9055 | 33.6098 | -42.5153 | 45.05** | 0.0000 | 0.6669 |
| Variation Income | -8.8710 | 23.5462 | -32.4172 | 23.13** | 0.0001 | 0.5015 |

** indicates a significance of less than 5%.

Source: Calculated by the authors with Stata14.

the clients with surplus capital and the clients that need capital for productive investments, and the roots of banking are, precisely, this activity [8, 14]. Therefore, our hypothesis H2 is confirmed.

We have also test that ethical banking has experienced a great growth, whilst conventional banking has reduced its average size. The mean differences of loans, funding, total assets and income variation are significant at a 1% significance level. Therefore, we can state that loans, funding, total assets and income have undergone a higher increase in ethical banking during the analyzed period, which is in line with GABV [8, 53] and confirms our hypothesis H3.

## Probit regressions

The probit regressions, which is shown in Table 9, between the banking model and the individual explanatory considered variables show similar results to the statistical test. Tier 1 is not related to banking model, but the other variables are significant at 1% significance level. Ethical banks are less indebted and have better guarantee and coverage ratio. They are more linked to real economy with greater loans to assets and funding to assets ratios. And, the growth is positively related to ethical banks. Hence, the three hypotheses have been validated. The joint consideration of all variables has not been possible due to the high correlation between the financial parameters.

## Discussion

The last financial crisis questioned the sustainability of conventional banks. Meanwhile the small and relatively new ethical banking overcame without too many problems such a crisis. The aim of this manuscript was to perform a financial and economic comparison between ethical and conventional banking, by pointing out the features of each kind of banking, specially

**Table 9. Probit regressions between banking model and the individual explanatory variables.**

| | Tier 1 | Indetebness | Guarantee | Coverage | Loans/Assets | Funding/Assets | Var.Loans | Var.Funding | Var.Assets | Var.Income |
|---|---|---|---|---|---|---|---|---|---|---|
| Constant | 0.3666 | 2.8638*** | -43.2372*** | -3.1618*** | -3.4145*** | -6.7201*** | -2.2852** | -0.5626 | -0.0243 | -0.0414 |
| | (0.604) | (0.000) | (0.000) | (0.000) | (0.000) | (0.000) | (0.045) | (0.388) | (0.975) | (0.945) |
| ROE | 0.0135 | 0.0025 | 0.0099 | 0.0091 | -0.0497* | 0.0739* | 0.1707 | -0.0973 | -0.1413 | -0.0567 |
| | (0.579) | (0.928) | (0.715) | (0.737) | (0.094) | (0.051) | (0.195) | (0.434) | (0.361) | (0.583) |
| Variable | -0.0178 | -0.1973*** | 40.2671*** | 46.2339*** | 6.2528*** | 9.4582*** | 0.1210*** | 0.1035*** | 0.1474*** | 0.0981*** |
| | (0.674) | (0.000) | (0.000) | (0.000) | (0.000) | (0.000) | (0.004) | (0.000) | (0.000) | (0.000) |
| Observations | 92 | 92 | 92 | 92 | 92 | 92 | 23 | 23 | 23 | 23 |
| Pseudo $R^2$ | 0.0047 | 0.2427 | 0.2720 | 0.2705 | 0.2216 | 0.5522 | 0.6725 | 0.6331 | 0.8003 | 0.5643 |
| McFadden's Adjust $R^2$ | -0.043 | 0.195 | 0.224 | 0.223 | 0.174 | 0.505 | 0.482 | 0.443 | 0.610 | 0.374 |
| Nagelkerke $R^2$ | 0.0085 | 0.3791 | 0.4170 | 0.4150 | 0.3510 | 0.7114 | 0.8070 | 0.7774 | 0.8928 | 0.7217 |
| Wald chi2 | 0.58 | 26.33*** | 29.59*** | 29.33*** | 32.94*** | 69.22*** | 8.79** | 15.44*** | 19.13 | 17.63*** |
| | (0.7489) | (0.0000) | (0.0001) | (0.0000) | (0.0000) | (0.0000) | (0.0124) | (0.0004) | (0.0001) | (0.0001) |
| Hosmer-Lemeshow chi2 | 18.20 | 15.68 | 14.85 | 14.99 | 9.04 | 13.78 | 1.54 | 6.75 | 0.56 | 2.63 |
| | (0.0198) | (0.0472) | (0.0621) | (0.0594) | (0.3390) | (0.0876) | (0.9920) | (0.5640) | (0.9998) | (0.9552) |
| Percent concordant | 56.52 | 70.65 | 72.83 | 72.83 | 67.39 | 80.43 | 91.30 | 86.96 | 91.30 | 78.26 |

*** and ** indicate less than 1% significance level and less than 5%, respectively.

*p*-values are shown into brackets.

According to Wald test *p*-value should be less than 0.05 and according to Hosmer-Lemeshow test *p*-value should be more than 0.05.

Source: Calculated by the authors with Stata14.

the liquidity, solvency and risk levels that shape the strength of ethical banking to face the future and to be a reliable banking for depositors.

Conventional banks are focused on profit maximization, whilst social banking applies the triple bottom-line of profit-people-planet. This is the main difference between the two banking models. Ethical banks are relatively small banks, particularly compared with conventional banks [5]. But, it is possible that the high social and environmental commitment only can be attained if the bank is small and flexible enough to make decisions quickly and to judge the borrowers based on a personal relationship [4].

Despite the loss of credibility in the banking system after the financial crisis, ethical banking has experienced a spectacular increase of funding, unlike conventional banking where they have decreased. This is a proof that depositors support sustainable banking [43]. The growth of funding could be a problem if there was no correspondence with the increase of loans, since savings should be invested in projects aimed to real economy and useful goals, such as social and environmental initiatives. However, in the analyzed sample both funding and loans have increased in a similar proportion, by 25%, which is an indicator of a balanced growth. Moreover, the fact that the increase of ethical banking is financed by client deposits implies that it is a kind of banking independent of financial and interbank markets [5] and, consequently, it is strong to remain stable in the face of economic fluctuations.

Sustainable banking does not aim a profit maximization, but it obviously needs to obtain profitability in its operations. We do not have found evidence, in the analyzed period and for the case of study Triodos and its competitors, that profitability is significantly different between conventional and ethical banking. On the contrary, two previous studies state that ethical banks experience less profit than conventional banks [5, 43]; however, the first one only includes two banks (Triodos and Banco Santander) to make the comparison [43] and the second one also includes alleviate-poverty banks [5], which perform different patterns. The GABV [8] states that ethical banks have resilient financial returns and this lower level of volatility leads to more stability in crisis periods, due to its link with the real economy and its rejection of toxic or complex financial products [10]. In fact, in our study we have verified that the variable "income" decreased more than 4% in conventional banking in the period 2015–2017, since it increased by 9.66% in the ethical banking, as a whole.

Regarding to the reliability for depositors, we have proved that, in the observed sample, ethical banking is undoubtedly surer than conventional banks, since the first one has a lesser indebtedness ratio and higher guarantee and coverage ratios. Therefore, investing in sustainable banking is safer [36]. Multiple reasons could be used to explain this fact. First, ethical banks base its activities on the real economy, by rejecting investment in structured and speculative products [8, 9]. Second, social banks are small and they are also specialized in specific sectors, which allows them to correctly assess the risk of the financed projects [14] and, consequently, they can keep a lesser default rate, as in the case of Triodos [37, 42]. In addition, it could exist reciprocity between the bank and the borrower; namely, banks give credits with advantaged terms for social and environmental projects and borrowers, who consider themselves fairly attended, respond with lesser default than conventional clients [36]. Third, customers choose sustainable banks for their ethics, which leads them to make more prudent decisions than conventional banking [14].

According to the above statements, ethical banking is a shining example to the financial sector overall [9], and it represents an attractive business case, which, however, it is not being adopted by conventional banks. This can be because of the inertia and the power of the established *status quo*, the cowardice of banking managers and shareholders to change the current model or even their limited awareness about data offered by works like this [8]. Ethical banking could provide important learning to the banking sector, in order to face future financial

crisis, since it is less speculative, more responsible and community and environmental oriented. However, current sustainable banks have a low market share within the banking and they cannot themselves cause a change in the global financial sector [4]. In order to exhibit an ethical behavior, formal rules and procedures are necessary, as well as new forms of social accounting, sustainability indicators and performance standards [30]. Moreover, given the small size and the domestic market of the current ethical banking overall, international associations as the GABV have a great interest because they provide some opportunities for mutual learning, to solve common problems and to influence policy making more effectively than individual banks [9].

If a company implemented unethical behavior, it could not survive in the long-term, since its reputation would be diminished and the financial sector is exposed, more than any other, to moral dangers [54]. Hence, the banking system will be strengthened through the growth of banks that operate in accordance with the principles of ethical banking and the risk of depositors will be reduced. Moreover, this kind of banking offers not only an economic, but also a social and environmental value to the stakeholders, namely, to shareholders, clients, employees, investors and the society as a whole [8].

To conclude, we can highlight the stability of sustainable banking in comparison to conventional banking, because the first one has experienced a great growth whilst the second one has decreased. Furthermore, ethical banking is based on real economy and not on speculative transactions, and presents a lower indebtedness ratio and higher guarantee and coverage ratio, overall. Therefore, ethical banking is less risky than conventional banks. Consequently, sustainable banking supposes compelling alternative to conventional banks, and provides a worthwhile precedent to achieve a new approach to sustainable finance.

In the future, it would be interesting to include a wide sample of ethical banking in the study of the financial statements, besides Triodos, and to expand the considered sample to a large number of conventional banks in Europe, analyse other continents, as well as a wider period and more variables.

## Supporting information

**S1 File. Dataset.** The "S1 File.Dataset" file includes the following information:

- "Conventional" sheet shows, for Triodos Bank and the 10 conventional banks, the following data corresponding to 2015–2018:

○ Country

○ Co-workers

○ Loans

○ Client funding

○ Non-current assets

○ Current assets

○ Equity

○ Non-current liabilities

○ Income

○ Expenses

○ Tases

○ TIER 1 capital

• "Ethical" sheet shows, for the 13 European ethical banks, the following data corresponding to 2015–2018:

○ Country

○ Total Assets

○ Loans

○ Client funding

○ Equity

○ Tier 1 capital

○ Total revenue

○ Net income

○ ROA

○ ROE

○ Co-workets

○ Clients (lending and deposit).
   (XLSX)

## Author Contributions

**Conceptualization:** María del Carmen Valls Martínez.

**Data curation:** María del Carmen Valls Martínez.

**Formal analysis:** María del Carmen Valls Martínez.

**Investigation:** María del Carmen Valls Martínez, Isabel María Parra Oller.

**Methodology:** María del Carmen Valls Martínez.

**Supervision:** Salvador Cruz Rambaud.

**Validation:** Isabel María Parra Oller.

**Writing – original draft:** María del Carmen Valls Martínez.

**Writing – review & editing:** Salvador Cruz Rambaud, Isabel María Parra Oller.

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
