## [Decision Letter · Decision Letter 0]

22 Nov 2019

PONE-D-19-27736

Sustainable and conventional banking in Europe

PLOS ONE

Dear Dr. Valls Martínez,

Thank you for submitting your manuscript to PLOS ONE. After careful consideration, we feel that it has merit but does not fully meet PLOS ONE’s publication criteria as it currently stands. Therefore, we invite you to submit a revised version of the manuscript that addresses the points raised during the review process.

I have received the reports from three experts in the field and all find merits in your manuscript. However, they also point out that some modifications should be carried out before accepting the manuscript for publication. I would like to highlight what I consider critical from their comments and some of my own regarding our publication policy.

1. Methods. R#1 suggests that you could use alternative methods that would add value to the manuscript. I would like to add that your current method does not allow to appropriately test your hypotheses.  That is because the ANOVA pools the data ending up with a cross-section. Therefore, there are a number of alternative explanations for your results that simply cannot be ruled out. The differences that you find could be attributed to the ethical attributes of the bank or any omitted variables that you left out the analysis. For more information on the validity of cross-sectional inferences, please see Spector, P. E. (2019). Do not cross me: optimizing the use of cross-sectional designs. *Journal of business and psychology*, *34*(2), 125-137.

I would suggest that you exploit the panel nature of your data and use panel fixed effects regression analysis to identify significant differences between conventional and non-convetional banks (with a dummy variable). This way, you can be more confident that the results are not driven by other characteristics or specific to particular point in time.

You could leave the ANOVA test as a robustness analysis.

2. Data. In order to perform the abovementioned analysis, please complete BBVA's international data (R#2) and try to expand your dataset to 2018 (R#1).

3. English. Please proof-read the paper before re-submitting it.

Additionally, address the minor changes and comments from the reviewers in the text.

We would appreciate receiving your revised manuscript by Jan 06 2020 11:59PM. To enhance the reproducibility of your results, we recommend that if applicable you deposit your laboratory protocols in protocols.io, where a protocol can be assigned its own identifier (DOI) such that it can be cited independently in the future. For instructions see: http://journals.plos.org/plosone/s/submission-guidelines#loc-laboratory-protocols

We look forward to receiving your revised manuscript.

Kind regards,

Jordi Paniagua

Academic Editor

PLOS ONE

Journal Requirements:

1.

We suggest you thoroughly copyedit your manuscript for language usage, spelling, and grammar. If you do not know anyone who can help you do this, you may wish to consider employing a professional scientific editing service.  

2. Please provide additional details in your Data Availability Statement regarding the dataset used in the analyses, such that other researchers could locate the data and replicate the analyses.

Reviewers' comments:

Reviewer's Responses to Questions

**Comments to the Author**

1. Is the manuscript technically sound, and do the data support the conclusions?

Reviewer #1: Partly

Reviewer #2: Yes

Reviewer #3: Yes

2. Has the statistical analysis been performed appropriately and rigorously? 

Reviewer #1: No

Reviewer #2: Yes

Reviewer #3: Yes

3. Have the authors made all data underlying the findings in their manuscript fully available?

Reviewer #1: Yes

Reviewer #2: Yes

Reviewer #3: No

4. Is the manuscript presented in an intelligible fashion and written in standard English?

Reviewer #1: Yes

Reviewer #2: Yes

Reviewer #3: Yes

5. Review Comments to the Author

Reviewer #1: The authors presents in this paper an interesting topic. The literatura review is correct and recent. However, regarding the study period I think it is reduced and could have been extended further. Only three years are studied from 2015 to 2017. When the year 2018 could have been incorporated, which already provides financial data of the entities object of this stusy. I think that the authors having provided information before and after the crisis that may be more interesting for the study.

Because in this way the time horizon of study is not very representative. At work it is not justified because the horizon under study from 2015 to 2017 is selected. And the relevance of this choice.

On the other hand, I miss a section of hypothesis to contrast. In this work the absence of hypothesis is a complication for the understanding of the results obtained. In my opinion there would have to be a hypothesis formulation section and subsequently its contrast in the results section or in another subsequent section.

The sample of European banks would expand it, I think it falls short. I leave it at the discretion of the authors

Regarding the methodology used, although valid for the work, another type of statistical methodology could have been used that would give more value to the work presented.

In this work it would also be interesting to provide the bibliographic reference of some previous work that uses the financial economic ratios used in this study. My apologies if it is but I can't find it.

With all this, I think it is relevant to expand the study horizon, it seems very short. And, on the other hand, to create a section of hypothesis formulation that allows a better understanding of the objectives pursued by this work and their contrast empirical.

Finally encourage the authors in this line of research that is interesting for the scientific community

Reviewer #2: This manuscript deals with ethical banking which is an interesting and current research topic, since ethical finance is an area of growing interest in the recent literature.

Authors analyze the concepts of socially responsible investment and corporate social responsibility in order to establish the basis of ethical banking and the differences between conventional and ethical banking. The two banking patterns are compared through their financial statements, by using data from 2015 to 2017. Authors conclude that conventional banking presents lower liquidity, guarantee and solvency than this new banking model, whilst profitability does not show remarkable differences. Authors compare the financial statements of Triodos Bank with ten conventional banks. Furthermore, they apply an ANOVA test to compare the European ethical banks and ten conventional banks from where they obtain interesting differences.

In my opinion, this contribution merits publication in the journal. However, the following theoretical issues must be more justified and elaborated in the text:

1. Authors consider that the financial crisis in 2008 led to a distrust towards conventional banking and, consequently, ethical banking emerged. However, ethical banking came up earlier to cover the needs of certain sectors of the population, specifically the so-called solidarity economy and third sector. Moreover, cooperative and savings banks have been ignored and they were born with similar objectives.

2. The article only consider two banking models, conventional and ethical banking. However, in several European countries there is also a public banking. Moreover, do cooperative and saving banks (which share characteristics with both types of banking) have to be considered as ethical or conventional banking? Why?

3. It would be convenient to present a table with the main differences between conventional and ethical banking.

4. Ethical banking presents not only a unique model. In effect, Triodos is an anonymous society but the cooperative model (for example, Banca Popolare Etica) has also to be taken into account. This issue has not been mentioned in the paper.

5. The data of BBVA correspond to Spain whilst the rest of entities provide international data. Therefore, data from BBVA must be completed in order to include international information.

6. Finally, I recommend a grammar revision of the text before its possible publication.

Reviewer #3: PONE-D-19-27736

Title: "Sustainable and conventional banking in Europe."

The aim of this paper is to analyze the financial and economic structure of ethical banking including a comparison with conventional banking in Europe, by considering the liquidity, solvency and coverage of both banking models. The results reveal that ethical banking presents greater liquidity, solvency and growth than conventional banking, but its profitability is not higher, in general terms. Thus, both investors and savers can be confident about their savings, since investments are made with ethical criteria and without additional risk. Moreover, whilst conventional banking carries out speculative transactions, ethical banking operates only in the real economy and presents a lower debt ratio and a higher coverage ratio. In conclusion, ethical banking exhibits less risk than conventional banks. Therefore, the ethical business model could be the answer to the current ethical financial problems.

I find this manuscript very interesting and timely, but there are some minor revisions, which should be addressed:

• Abstract

o It is stated that “The results reveal that ethical banking is growing more than conventional banking and it presents greater liquidity and solvency, although, in general terms, its profitability is lower”. Additionally, in Section 3.3, it is stated that “Regarding to ROA and ROE, we do not find a common pattern in conventional banks and different in Triodos”. Consequently, I propose that, in the abstract, authors should state: “The results reveal that ethical banking is growing more than conventional banking and it presents greater liquidity and solvency, although, in general terms, its profitability is not higher”.

o Authors should mention the mean difference test and the analysis of variance.

• Introduction

o The authors use the terms ethical banking and social banking in the same sense when it is clear that both concepts are different. The difference between them should be specified. This is because cooperative banks and saving banks have been considered as social banks and their differentiation should also be considered.

o In the fifth paragraph, it is stated: “First, we van highlight the distrust…”. It is clear that there is a typo which must be corrected: “First, we can highlight the distrust…”.

o Table 1 should provide the information corresponding to 2016, in line with the rest of the tables.

• Discussion

o Authors finish conclusions by stating that “It would be interesting to extend in the future the considered sample to a large number of conventional banks in Europe and even to analyse other continents, as well as a wider period and more variables”. I consider that it would be interesting to include a wide sample of ethical banking in the study of the financial statements and not only Triodos.

• References

o The references Cheong et al. (2016) and Kant et al. (2017) are included in the text, but they are not included in the reference list.

Finally, the paper has some grammar mistakes and I would suggest the author(s) proofread the paper, if possible.

6. PLOS authors have the option to publish the peer review history of their article (what does this mean?). If published, this will include your full peer review and any attached files.

Reviewer #1: No

Reviewer #2: No

Reviewer #3: No

---

## [Decision Letter · Decision Letter 1]

6 Feb 2020

Sustainable and conventional banking in Europe

PONE-D-19-27736R1

Dear Dr. Valls Martínez,

We are pleased to inform you that your manuscript has been judged scientifically suitable for publication and will be formally accepted for publication once it complies with all outstanding technical requirements.

With kind regards,

Jordi Paniagua

Academic Editor

PLOS ONE

Additional Editor Comments (optional):

Reviewers' comments:

Reviewer's Responses to Questions

**Comments to the Author**

1. If the authors have adequately addressed your comments raised in a previous round of review and you feel that this manuscript is now acceptable for publication, you may indicate that here to bypass the “Comments to the Author” section, enter your conflict of interest statement in the “Confidential to Editor” section, and submit your "Accept" recommendation.

Reviewer #1: All comments have been addressed

Reviewer #2: All comments have been addressed

Reviewer #3: All comments have been addressed

2. Is the manuscript technically sound, and do the data support the conclusions?

Reviewer #1: Yes

Reviewer #2: Yes

Reviewer #3: Yes

3. Has the statistical analysis been performed appropriately and rigorously? 

Reviewer #1: Yes

Reviewer #2: Yes

Reviewer #3: Yes

4. Have the authors made all data underlying the findings in their manuscript fully available?

Reviewer #1: Yes

Reviewer #2: Yes

Reviewer #3: No

5. Is the manuscript presented in an intelligible fashion and written in standard English?

Reviewer #1: Yes

Reviewer #2: Yes

Reviewer #3: Yes

6. Review Comments to the Author

Reviewer #1: Congratulations to the authors, the study presents the results of original research. Thus, statistics analyses are performed to a high technical standard and are described in sufficient detail. Also, conclusions are presented in an appropriate fashion and are supported by the data. The article is presented in an intelligible fashion and is written in standard English. The research meets all applicable standards for the ethics of experimentation and research integrity.

The article adheres to appropriate reporting guidelines and community standards for data availability.

With all this, my recomedation is accept the research for the publicacion of the paper.

Reviewer #2: The authors have addressed all my concerns. Moreover, they have improved the paper also regarding the English language and the readability. In my opinion, the paper is ready to be published.

Reviewer #3: The authors have perfectly taken my recommendations into account. I think this is a very interesting paper, really worth publishing.

7. PLOS authors have the option to publish the peer review history of their article (what does this mean?). If published, this will include your full peer review and any attached files.

Reviewer #1: Yes: Dr. D. Miguel Ángel Latorre Guillem

Profesor de la Faculta de Ciencias Jurídicas, Económicas y Sociales

Universidad Católica de Valencia "San Vicente Mártir"

mangel.latorre@ucv.es

Reviewer #2: No

Reviewer #3: No

---

## [Editor Report · Acceptance letter]

12 Feb 2020

PONE-D-19-27736R1 

Sustainable and conventional banking in Europe 

Dear Dr. Valls Martínez:

I am pleased to inform you that your manuscript has been deemed suitable for publication in PLOS ONE. Congratulations! Your manuscript is now with our production department. 

With kind regards,

on behalf of

Dr. Jordi Paniagua 

Academic Editor

PLOS ONE